# Environmental remodeling of human gut microbiota and antibiotic resistome in livestock farms

Jian Sun[1,2,10], Xiao-Ping Liao [1,2,10], Alaric W. D'Souza [3,10], Manish Boolchandani [3,10], Sheng-Hui Li[1,4,10], Ke Cheng[1,2], José Luis Martínez [5], Liang Li[1,2], You-Jun Feng[1,2], Liang-Xing Fang[1,2], Ting Huang[1,2], Jing Xia[1,2], Yang Yu[1,2], Yu-Feng Zhou[1,2], Yong-Xue Sun[1,2,6], Xian-Bo Deng[2], Zhen-Ling Zeng[1,2,6], Hong-Xia Jiang[1,2,6], Bing-Hu Fang[1,2,6], You-Zhi Tang[1,2,6], Xin-Lei Lian[1,2], Rong-Min Zhang[1,2], Zhi-Wei Fang[4], Qiu-Long Yan[4], Gautam Dantas [3,7,8,9✉] & Ya-Hong Liu [1,2,6✉]

Anthropogenic environments have been implicated in enrichment and exchange of antibiotic resistance genes and bacteria. Here we study the impact of confined and controlled swine farm environments on temporal changes in the gut microbiome and resistome of veterinary students with occupational exposure for 3 months. By analyzing 16S rRNA and whole metagenome shotgun sequencing data in tandem with culture-based methods, we show that farm exposure shapes the gut microbiome of students, resulting in enrichment of potentially pathogenic taxa and antimicrobial resistance genes. Comparison of students' gut microbiomes and resistomes to farm workers' and environmental samples revealed extensive sharing of resistance genes and bacteria following exposure and after three months of their visit. Notably, antibiotic resistance genes were found in similar genetic contexts in student samples and farm environmental samples. Dynamic Bayesian network modeling predicted that the observed changes partially reverse over a 4-6 month period. Our results indicate that acute changes in a human's living environment can persistently shape their gut microbiota and antibiotic resistome.

[1] National Risk Assessment Laboratory for Antimicrobial Resistance of Animal Original Bacteria, South China Agricultural University, Guangzhou, Guangdong, China. [2] Guangdong Laboratory for Lingnan Modern Agriculture, Guangzhou, Guangdong, China. [3] The Edison Family Center for Genome Sciences and Systems Biology, Washington University in St. Louis School of Medicine, St. Louis, MO, USA. [4] Shenzhen Puensum Genetech Institute, Shenzhen, Guangdong, China. [5] Centro Nacional de Biotecnología, Consejo Superior de Investigaciones Científicas, Calle Darwin, Madrid, Spain. [6] Guangdong Provincial Key Laboratory of Veterinary Pharmaceutics Development and Safety Evaluation, South China Agricultural University, Guangzhou, Guangdong, China. [7] Department of Pathology & Immunology, Washington University in St. Louis School of Medicine, St. Louis, MO, USA. [8] Department of Biomedical Engineering, Washington University in St. Louis, St. Louis, MO, USA. [9] Department of Molecular Microbiology, Washington University in St. Louis School of Medicine, St. Louis, MO, USA. [10]These authors contributed equally: Jian Sun, Xiao-Ping Liao, Alaric W. D'Souza, Manish Boolchandani, Sheng-Hui Li. ✉email: dantas@wustl.edu; lyh@scau.edu.cn

The human gut microbiota is a dynamic ecosystem of commensal microbes which collectively modulate host health and physiology[1,2]. Previous studies have revealed that the human gut microbiota composition is stable over time[3–5] and to some extent resilient to short-term perturbations[6,7]. While the host gut microbiota is generally predicted to recover to pre-perturbation states[8], the exact extent of reversion and stability of different human microbiomes subjected to different types of environmental perturbations remains underdetermined.

The human gut microbiota composition is generally influenced by both host genetics and environment[9–11]; however, the effect of environmental factors appears to outweigh host genetics in shaping the microbiota[12]. Indeed, changes in diet[13], geography[14,15], and chemotherapeutics[16–18] (e.g., antibiotics) have been shown to rapidly alter microbiota composition and affect colonization resistance to pathogens[19]. A recent study demonstrated that environments harbor microbial communities which can serve as hot-spots of resistance gene enrichment and exchange[20]. One such environment is swine farms where antibiotics are administered routinely for growth promotion and disease prevention in major food producers such as China[21], providing ideal selection pressure for enrichment of antibiotic-resistant bacteria and antibiotic resistance (AR) genes. Indeed, mcr-1, a plasmid-borne resistance gene against colistin, a drug of last-resort, was first reported in 2016 at a Chinese swine farm[22]. Direct evidence has shown environmental transmission of AR genes and their bacterial hosts among livestock and humans[23]. In addition, these antibiotic-resistant bacteria and AR genes can spread to humans via contaminated meat, pig-house dust and manure, and wastewater discharge[24–26]. Interacting with swine farm environments where antibiotic use is prevalent has been considered a potential high-risk factor for infection with multidrug-resistant bacteria[27]. The influence of antibiotic use on human health is dependent on the connectivity between the farms and human-associated microbiomes. This connectivity includes both the transmissibility of antibiotic-resistant bacteria selected in animals to human hosts, as well as the potential of lateral AR gene transfer between animal-associated and human-associated bacteria. It is critical not only to determine the extent of human microbiota disruption in such environmental exposures, but also the distribution and enrichment of AR genes to evaluate the potential risks of these environments in facilitating the global dissemination of AR[28–30].

Here, we report a longitudinal investigation of confined and controlled swine farm environments impacts on temporal changes in the gut microbiome and resistome of 14 healthy students who underwent occupational exposure during 3-month internships at swine farms. Compared to their pre-internship baseline (T0), we found that the students' gut microbiotas became more similar in composition to full-time farm workers' gut microbiotas in correlation with swine farm environmental exposure, that student and swine farm environmental microbiota and resistome appear extensively interconnected following exposure, and that changes in students' gut microbiota community structure partially reverted 6 months after they returned home. This study presents insights on how, and to what extent, temporary changes in living environments can shape the human gut microbiota and resistome.

## Results

**Gut microbiota changes correlate with environmental exposure.** We performed 16S rRNA gene sequencing of 91 fecal samples collected longitudinally from 14 male student volunteers (at time points T0, T1–T3, and T4–T6; Fig. 1a), randomly assigned to different large-scale farms in China (Supplementary Fig. 1; Supplementary Data 1), to characterize temporal patterns in gut microbial community structure that occur with environmental changes. Multivariate analysis of operational taxonomic unit (OTU) composition revealed a modest yet significant change ($R^2 = 7.4\%$, permutational multivariate analysis of variance [PERMANOVA] $P < 0.001$) in the gut microbial communities of the study participants over the period from swine farm arrival (T0) to leaving the farm environment (Fig. 1b). This change occurred within 1 month (T1) of the students reaching the swine farms. Three months after leaving the swine farms (T6), the students' gut microbiota partially reverted to their original microbial composition. Notably, the students' gut microbiota changed in a similar fashion at all three farms, likely reflecting the commonalities of the farm ecological environment despite geographical separation (Supplementary Fig. 2). The microbial diversity (alpha diversity) within the subjects' gut microbiota did not significantly differ (pairwise Student's t-test with Benjamini–Hochberg correction, $q > 0.05$; Fig. 1c) during their swine farm residence. However, granular analysis of specific microbial taxa showed marked deviation between students' arrival at the swine farm and their return to school. Specifically, we observed a moderate decrease in Bacteroidetes (major symbionts in the human gut that contribute to dietary carbohydrate metabolism and vitamin biosynthesis[31]) and an increase in Proteobacteria (especially Gammaproteobacteria, which includes many human pathogens) (Supplementary Fig. 3a; Supplementary Data 2), as well as significant changes in the relative abundance of several taxa from the *Faecalibacterium*, *Collinsella*, and *Blautia* genera, and the *Veillonellaceae* family (Supplementary Fig. 3b).

To further investigate the extent of alteration in students' gut microbiota, we performed whole-metagenome shotgun sequencing (WGS) on 42 fecal samples of students at time points T0, T3, and T6, and on fecal samples of three full-time workers from each swine farm (representing 336.9 Gb of high-quality data; Supplementary Data 3). Distance-based redundancy analysis (dbRDA) of microbial taxa (Supplementary Fig. 4a, b) showed marked deviation of students' gut microbiota at T3 from the pre-exposure time-point (T0), and similarity to the swine farm workers' microbiota. These results demonstrate that working in the swine farm environment is correlated with alterations in the visiting students' gut microbiota to more closely resemble the full-time workers' gut microbiota. These results were supported by significant higher Bray–Curtis dissimilarity between the T0 and T3 collections compared to the T0 and T6 collections (Supplementary Fig. 4c). To reduce the potential effect of time-dependent confounding factors, we compared the gut microbiomes of our participants with a baseline healthy cohort of 196 urban Chinese subjects collected across all seasons from the urban Chinese environment[32]. Our results further indicated that the T3 collection was significantly more dissimilar from controls than either the T0 or T6 times (Supplementary Fig. 4d). Since many environmental factors, including diet[13,33], antibiotics[18], and geography[15], have been associated with changes to the human gut microbiota, it is difficult to identify specific contributions from separate factors from the farm environment[34]. However, we observed the same trends of microbial community shift in all the students despite individual host and geographical location differences (Supplementary Fig. 4e), mirroring the taxonomic trends from the 16S rRNA gene-based analysis. These results suggest multiple conserved environmental factors on the swine farms from divergent geographic locations can consistently shape the gut microbiota.

**Antibiotic resistome structure influenced by changing environment.** To evaluate whether AR gene changes accompanied the microbiota changes, we performed metagenomic analysis of the students' gut resistomes. We identified 1924 non-redundant AR

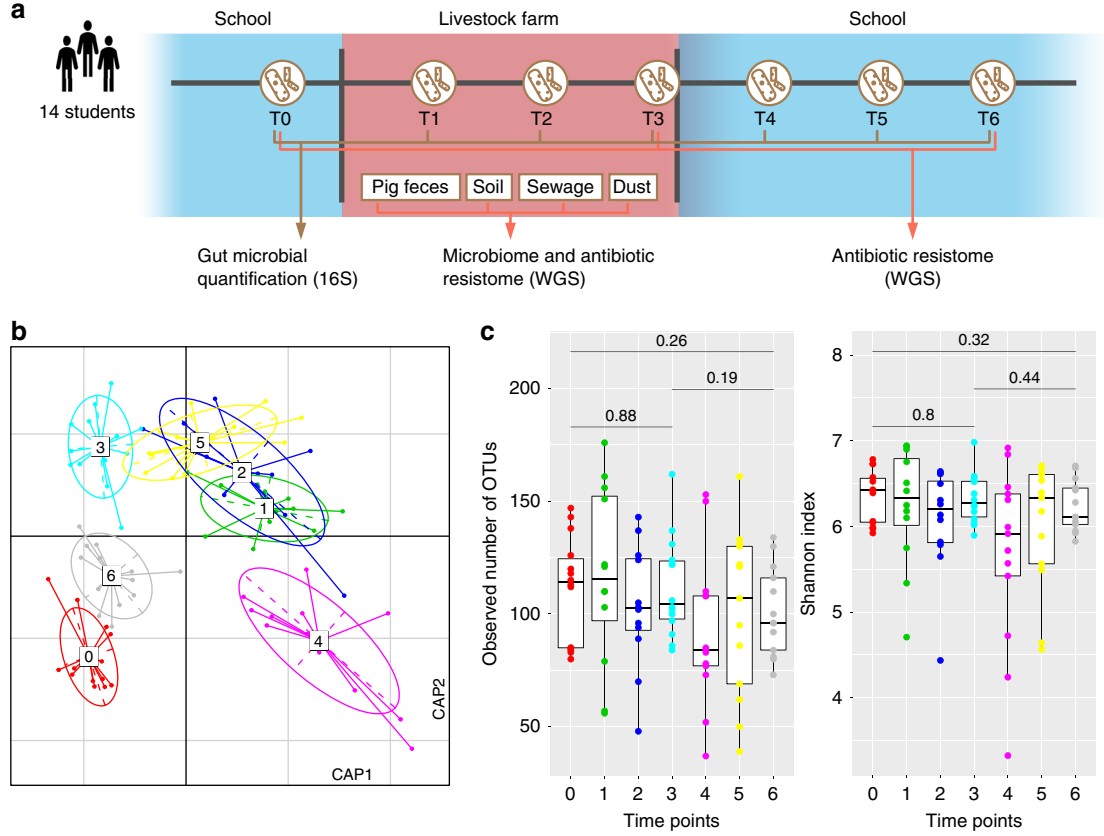

**Fig. 1 Change in the human gut microbiota following environmental conversion. a** Overview of the study design. Fourteen veterinary students' fecal samples were collected at seven time points: T0, 1–2 weeks before work on the swine farm; T1–T3, while living and working at the swine farm; T4–T6, after returning to the university. **b** Distance-based redundancy analysis (dbRDA) revealed gut microbiota dysbiosis during the students' swine farm stays, which partially recovered after leaving the farm. dbRDA of Bray–Curtis distances between operational taxonomic units (OTUs, based on 16S sequences) in samples at all-time points is shown at the first two constrained principal coordinates (CAP1 1.8% variance explained, CAP2 1.3% variance explained). Lines connect samples from the same time point, and colored circles indicate the samples near the center of gravity for each time point. The results depicted here are cumulative of the samples from three swine farms. **c** Change in the within-sample microbial diversity (observed number of OTUs and Shannon diversity index) of samples at seven different time points. Boxes show the distribution of students' samples (*n* = 14 biologically independent samples per timepoint) (boxes show medians/quartiles; error bars extend to the most extreme values within 1.5 interquartile ranges). *P* > 0.05 by Student's *t*-tests (paired two-sided test between the students' samples at time points 0, 3, and 6). *P*-values are multiple hypothesis test corrected using Benjamini–Hochberg (FDR) correction. Underlying data are provided in the Source Data file.

genes in the WGS samples (Supplementary Data 4). These genes encoded a range of AR enzymes, with beta-lactamases (44.4%), aminoglycoside resistance proteins (17.5%), and chloramphenicol acetyltransferases (15.2%) representing the most dominant types.

The abundance of AR genes in the students' samples was quantified using ShortBRED[35] based on a custom AR gene database that included unique protein markers created from Comprehensive Antibiotic Resistance Database (CARD ver. 2.0.0)[36] and AR genes we identified using metagenomic assembly. Furthermore, alterations in the visiting students' gut resistomes was different from the full-time workers' gut resistomes. However, similar to the gut microbiota composition changes, the students' gut resistome showed minor divergence between the samples taken at the three swine farm-stay time points (PERMANOVA *P* = 0.63 among three time points; Fig. 2a). Procrustes analysis confirmed that antibiotic resistomes were significantly correlated with community composition (PROTEST *P* < 0.001; Fig. 2b). There was no significant change in the number and abundance of AR genes detected in the samples obtained during the swine farm residence period (Fig. 2c; Supplementary Fig. 5a, b), despite an average 3.7% increase in relative abundance (measured by normalized reads per kilobase per million reads, RPKM). This increase was observed for several AR mechanisms (Supplementary Fig. 5c, d) including in subclass B3

β-lactamase, aminoglycoside acetyltransferase, and tetracycline-resistant ribosomal protection proteins. Interestingly, these resistome changes occurred after only 3 months in the swine farm environment.

**Microbial transmission from the swine farm environment into students' gut.** To assess whether changes in the students' gut microbiota and antibiotic resistome were correlated with their swine farm environmental contact, we examined the microbial landscape of the swine farm ecosystem via four representative environments: ventilation system dust, swine feces, sewage, and compost soil. Pooled environmental samples (each environment contained 3–5 sampling sites) were collected from each swine farm and analyzed using WGS (representing 133.2 Gb of data; Supplementary Data 5). The presence of many non-redundant genes in these habitats revealed the magnitude of diversity of the swine farm ecosystem (Supplementary Fig. 6). Compared to the human gut microbiome, the environmental samples exhibited higher microbial taxonomic and AR gene diversity (Supplementary Fig. 7; Supplementary Notes).

Comparison of student microbiota samples before and during swine farm residence revealed a high proportion of new genes (average increase of 42%, range 18–61%) identified after they

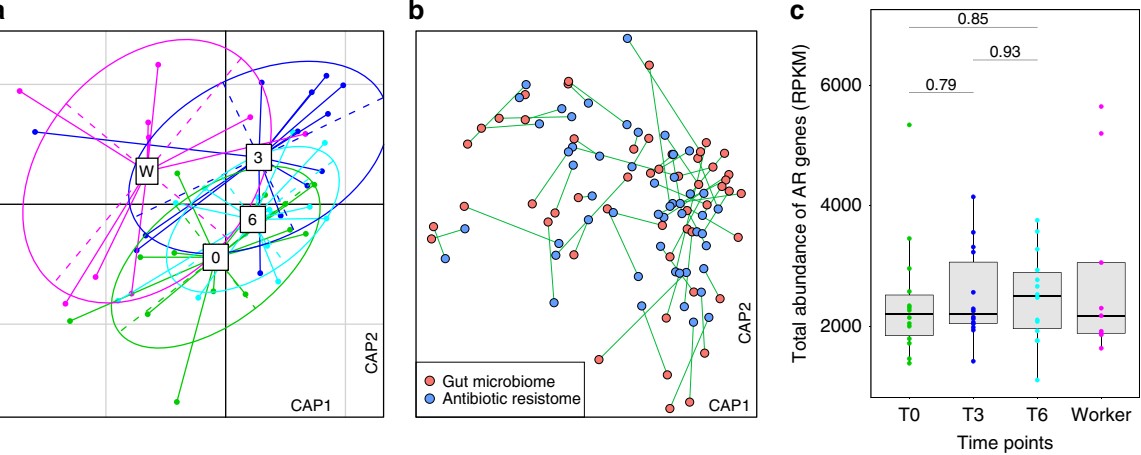

**Fig. 2 Change in the gut antibiotic resistomes following environmental exposure. a** Distance-based redundancy analysis (dbRDA) plot of the gut antibiotic resistomes of students' samples at time points 0 (red), 3 (green) and 6 (blue), and workers' samples. Lines connect samples from the same time point, and colored circles indicate the samples near the center of gravity for each time point. The first two constrained principal coordinates are shown (CAP1 2.8% variance explained, CAP2 1.6% variance explained). **b** Procrustes analysis connecting the microbiomes and resistomes of gut microbiota. **c** total abundance of the antibiotic resistance genes in students' samples at time points 0, 3, and 6, and workers' samples. Box plots show the distribution of students' samples ($n = 14$ biologically independent samples per timepoint) (boxes show medians/quartiles; error bars extend to the most extreme values within 1.5 interquartile ranges). $P > 0.05$; Student's $t$-tests (paired two-sided test between the students' samples at time points 0, 3, and 6). $P$-values are multiple hypothesis test corrected using Benjamini–Hochberg (FDR) correction. Underlying data are provided in the Source Data file.

arrived at the swine farms. Nearly two-thirds of these genes were also present in microbiomes derived from the environmental samples (Fig. 3a; Supplementary Fig. 8). Since genes can transfer via host bacterial transmission or lateral gene exchange, we sought to correlate our gene exchange results with the OTU composition change of the students during their swine farm stays. Strong association between the OTU composition changes and gene exchange would suggest extensive microbial transmission between the environment and the human gut. The SourceTracker algorithm[37] was used to interrogate this question by integrating the taxonomic assignments and the abundance levels of the newly acquired genes. One hundred and forty-two species transmission events were identified from various swine farm environments to the students' gut microbiota (Fig. 3b; Supplementary Fig. 9; Supplementary Data 6), and swine feces and soil were the main bacterial sources. These transmission events included diverse groups of Firmicutes and Proteobacteria, some of which (e.g., *Ruminococcus* spp., *Escherichia* spp., and *Pseudomonas putida*) include pathogenic strains responsible for zoonotic infections. Transmission of these commonly pathogenic species indicates that the soil and swine feces may be underappreciated occupational hazards of industrialized farming. To confirm that the species and their genes were environmentally acquired, we performed comparative genome analysis on these putatively transmitted species identified from the student and the environment microbiomes. Draft genomes of 9 high-abundance species, including the genome of *Phascolarctobacterium succinatutens*, which is rarely observed in the human gut, were reconstructed from the students' gut microbiomes and from the corresponding environmental samples (see the "Methods" section). These genomes shared 99.9 ± 0.1% (minimum 99.7%) 16S rRNA gene similarity and 99.5 ± 0.4% (minimum 98.9%) average nucleotide identity (ANI) with their respective environmentally derived genomes (Fig. 3c; Supplementary Data 7), suggesting that they belong to the same bacterial clones shared by the students and their surrounding environments.

To further study microbe transmission between students and farm environment/workers, we cultured and characterized the genetic relatedness of 82 *E. coli* strains isolated from students,

farm workers, and environment samples collected from one pig farm. We found multiple events of clonal spread of *E. coli* strains between students, farm workers, and farm environment (Supplementary Fig. 10). Together, the results from culture-independent analysis of fecal metagenomes and from culture-dependent analysis of environmental bacterial clones revealed extensive transmission events between the students' gut microbiome and farm environment occurred for diverse taxonomic groups, including putatively pathogenic bacteria.

Gene content from the 142 putatively transmitted species was further analyzed to identify genes with putative clinical relevance (e.g., chromosome-encoded AR genes and virulence factors), likely to transfer concomitantly with the microbes. Approximately 27% of predicted species transmission events between the environment and human gut microbiota carried at least one AR gene on their contigs (Supplementary Fig. 11a). *S. marcescens* strains carried the largest number of AR genes (average 11 genes), though this may reflect their high assembly completeness in our dataset. Additionally, many genes encoding virulence factors (observed in 30% of species transmission events), antibacterial biocide resistance genes (18%), and heavy metal resistance genes (18%) were also concomitantly transferred (Supplementary Fig. 11b–d). These results agree with our prior Procrustes analysis and demonstrate comprehensive accompanying transfer of clinically relevant genes with environment-mediated microbial transmission events. Altogether our results suggest an extensive exchange of bacteria (including pathogens and antibiotic-resistant bacteria) between humans and their surrounding environments.

**Transfer of AR genes between the swine farm environment and human gut.** The environment represents an enormous reservoir of antibiotic-resistant bacteria and AR genes[38], and its transmissibility to humans is concerning[20]. In our dataset, we found an extensive network of AR gene sharing between microbial communities of humans and environments (Supplementary Fig. 12a, a similar network is found in ref. [20]), and further revealed that 25% (477/1924) of AR genes detected in the students' microbiota while on the farms co-localized with putative mobile genetic elements (MGEs), which are often involved in AR transfer across

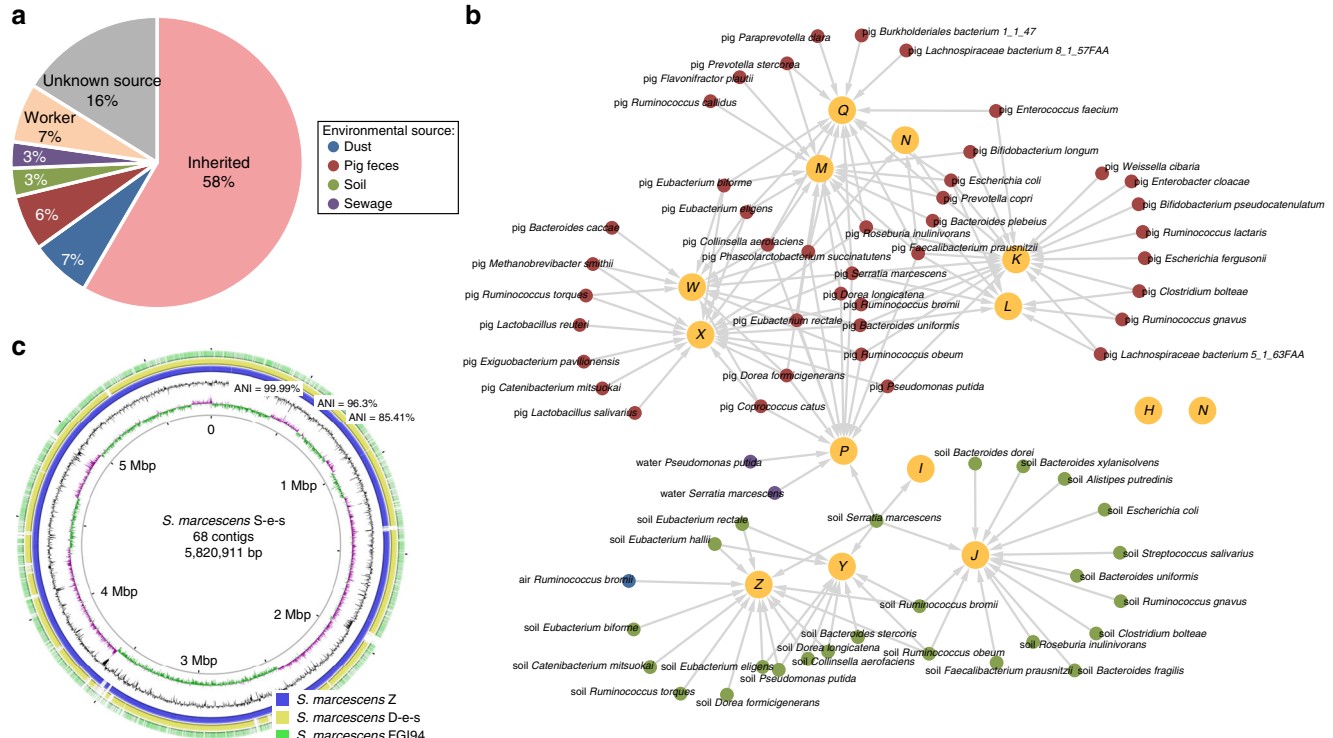

**Fig. 3 Transmission of microbes and antibiotic resistance genes from swine farm environments. a** Origin of the genes observed in the students' gut microbiomes during their stay on the swine farms (time point 3). The majority of the genes did not change (58%), but a large number of the newly observed genes were also identified in various swine farm habitats, such as the environment (19%, including dust, pig feces, soil, and sewage) and the workers' gut microbiota (7%). **b** Species transmission network from the swine farm environment to the human gut. Larger nodes depict the students (student ID is displayed in the center of each node). Smaller nodes depict transmitted species, color-coded according to environmental types. Connecting arrows represent the transmission events. **c** Circular representation of the *S. marcescens* S-e-s draft genome (assembled from a soil sample from swine farm S) and comparison to other genomes. The inner three circles represent the genome scale, G + C skew and G + C content of the S-e-s draft genome. The outer three circles show the portions of the S-e-s genomic region that have close orthologs in other draft genomes: *S. marcescens* Z (blue, from one student who inherited S-e-s), *S. marcescens* D-e-s (yellow, from the soil sample taken from swine farm D) and *S. marcescens* FGI94 (green, the most homologous genome from the NCBI database). High ANI (99.9%) between *S. marcescens* S-e-s and Z confirmed the inheritance relationship between them. Underlying data are provided in the Source Data file.

environments. Using the SourceTracker algorithm, we identified 270 AR genes involved in transfer from swine farm environmental samples to the students' gut microbiotas (Supplementary Fig. 12b; Supplementary Data 8). Swine feces and sewage were the major antibiotic-resistant bacteria and AR gene sources; transfer events from these sources included almost all types of AR genes, whereas soil mainly contributed the transfer of AR genes encoding aminoglycoside-inactivating enzymes. Of note, several studies have confirmed that soil[39], sewage[40], and even air dust[41] are likely significant reservoirs involved in spreading AR genes persistently found in clinical pathogens.

To further link the emergence of AR genes conferring resistance to medically important antimicrobials with their acquisition from the pig farm environment, we identified 120 relevant AR genes. These genes included extended-spectrum β-lactamases (including $bla_{TEM}$ and $bla_{CTX-M}$), the plasmid-mediated quinolone resistance genes *qnrS*, and the tigecycline resistance gene *tet(X)*. These AR genes were enriched in the students' gut resistome during or after swine farm stays (Fig. 4a; Supplementary Fig. 13). To understand the exchange potential of these AR genes, we examined the flanking genetic sequences in assembled contigs. 41% (49/120) of genes encoded by both human gut and environmental microbiota were found in consistently similar genetic contexts in the two habitats, and many of those were associated with MGEs (including several extended-spectrum β-lactamases, *tet(X)*, and *qnrS*; Fig. 4b shows

an example of $bla_{CTX-M}$). This provides evidence for sharing of important AR bacteria and AR genes between the human gut and swine farm environment microbiota.

To further investigate the acquisition of AR genes and phenotypic AR during the students' stay on the farms, we isolated 1851 *E. coli* strains from all samples. Phenotypic resistance testing showed that the resistance rates to nine antibiotics including cefotaxime, ciprofloxacin, and fosfomycin, increased among *E. coli* strains from students' samples at time points T1, T2, and T3 compared to time point T0. Interestingly, these resistance rates maintained high levels among *E. coli* strains from students' samples at post-return time points T4, T5, and T6 (Supplementary Fig. 14). This is consistent with metagenomic analyses that the resistome at T6 did not move towards the T0 state with the microbiota but instead retained high AR gene abundance. Notably, relatively high resistance rates to these drugs were found among *E. coli* strains from farm samples including farm workers, pigs, and environmental samples. Consistent with these phenotypic AR results, the detection rate of transferable plasmid-mediated AR genes, $bla_{CTX-M}$ (conferring resistance to third generation cephlosporins) and *fosA3* (conferring resistance to fosfomycin), also increased among *E. coli* strains from students during T2–T4 compared to T0. These rates declined during T4–T6 (Fig. 4c, d). We found *fosA3* genes co-localized with $bla_{CTX-M-14}$ on an identical genetic structure ($bla_{CTX-M-14}$-△IS*903*-261bp-*fosA3-orf1-orf2*-IS*26*) among 9 of 15 *E. coli* strains co-carrying $bla_{CTX-M-14}$ and

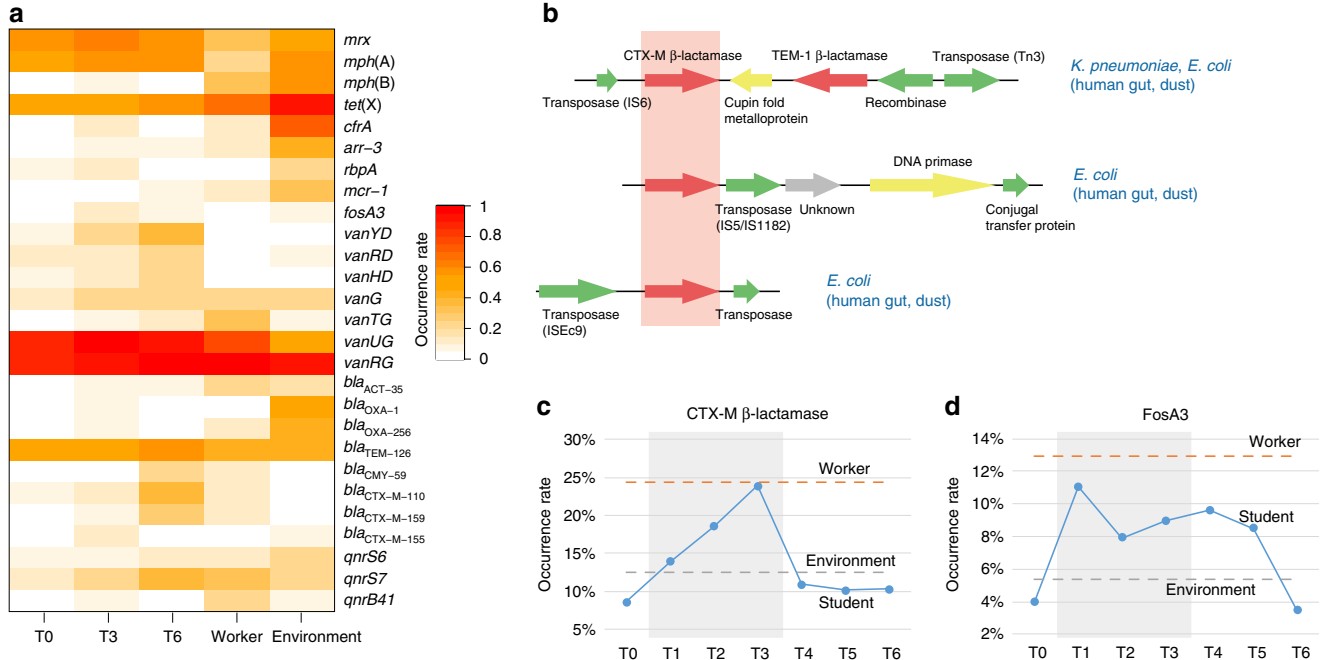

**Fig. 4 Accumulation of important antibiotic resistance genes in the human gut. a** Occurrence of the important antibiotic resistance genes in the microbiota of the students, the swine farm workers and the environment. Only important antibiotic resistance genes that were enriched in students' gut antibiotic resistomes during their swine farm stays (time point T3) are shown. The occurrence rates of antibiotic resistance genes in each group are represented by color shading. **b** Representative alignment of three contigs encoding a CTX-M β-lactamase with 99% nucleotide identity. The taxonomic assignments of the contigs are indicated on the right, and source metagenomic libraries are indicated inside the parenthesis. **c** and **d** Changes in resistance detection rate of the $bla_{CTX-M}$ gene **c** and *fosA3* gene **d** among 1851 *E. coli* strains from students (blue), farm workers, and environment during the students' swine farm residence period. The dotted lines for worker and environmental samples are the average occurrence rate of resistance genes in *E. coli* strains from farm workers (orange) and environment (gray) in T1, T2, and T3. Underlying data are provided in the Source Data file.

*fosA3* from students ($n = 4$), pigs ($n = 4$), and compost soil ($n = 1$). These results indicate that the pig farm environment probably contributes to AR increases in human commensal *E. coli* isolates through the transmission of AR *E. coli* isolates or AR gene transfer.

**Predicting the impact of environmentally induced changes on the gut microbial community structure**. To predict the duration of the effect on the gut microbial community structure by environmental changes, we developed a time series model of relative taxonomic abundance based on the 16S sequencing data obtained from the students' fecal samples at seven time points (see Online Methods). We modeled microbial interactions as a dynamic Bayesian network (DBN) using extended local similarity analysis, to capture local and potentially time-delayed co-occurrence and association patterns between microbial taxa (Supplementary Fig. 15). Analysis of the extrapolated community structure showed that the students' gut microbiotas are likely to revert to the original status within 4–6 months of returning to their initial environment (Fig. 5). The change in taxonomic community structure stalls and stabilizes in the 6–9-month range. The time point at 9 months begins to have a wider distribution as it is further removed from the last measured sample. As the gut resistome and gut microbiota phylogenetic composition appeared to be tightly linked, we also found resistome reversion after terminating the exposure to the swine farm environment. However, some clinically relevant AR genes were persistently isolated after a 3-month recovery period. To elucidate this phenomenon a more extensive longitudinal study of environmental shaping of the human gut microbiota and resistome along with subsequent recovery is required.

**Discussion**

Environmental factors influence human health[10]. One mechanism for this influence is interaction between environmental and human-associated microbiotas[12]. In this study, we used both endpoint and time series analyses to demonstrate that the human gut microbiota and resistome undergoes taxonomic and functional remodeling in correlation to exposure to the high-risk swine farm environment. We found substantial interconnection of microbiomes and resistomes between the swine farm environment and visiting veterinary students, and the diversity of source environments we observed in our finite sampling may still only represent a fraction of the true bacterial reservoirs. The environment microbiome harbors a diversity of the microorganisms, and humans may interact with these microbes via direct or indirect contact during environmental microbial exposures[12]. This is especially important if humans may acquire new commensal species or dangerous pathogens from environment. Although the volunteer veterinary student population is a small sample size and only involved in male participants, our results provide direct evidence that the human gut microbiota can change in response to environmental conversion. These environmental conversions likely work in concert with other factors, such as age, biological sex, personal hygiene, dietary habits, antibiotic use, and stress to shape the microbiome. These acute changes may considerably impact human health and could represent underappreciated occupational hazards. Future studies should seek to clarify the key roles of reservoirs, carriers, and vectors on the transmission chain and to identify factors promoting AR gene exchange between environmental microbiota and human commensal bacteria. Thus, a quantitative model for assessing resistance gene transmission risk to humans is urgently needed.

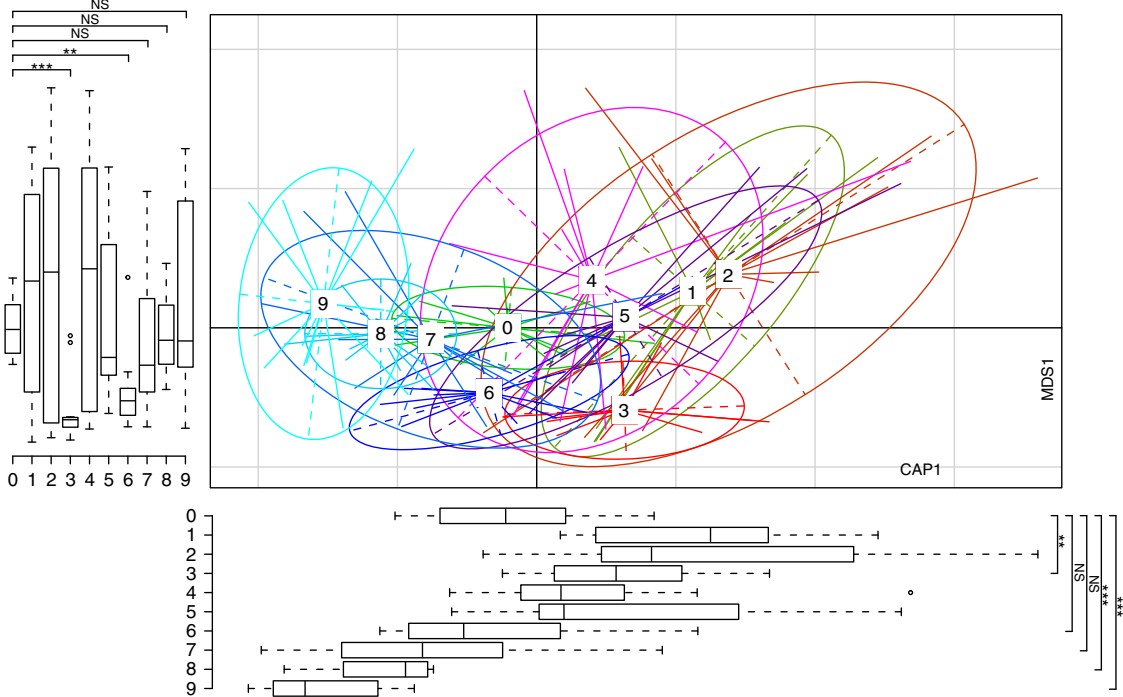

**Fig. 5 Predicting students' gut microbiotas in the next 3 months using a dynamic Bayesian network model.** DbRDA of the Bray–Curtis PCoA of the unweighted UniFrac distances between the gut microbiota in samples at the seven time points tested and three predicted future time points. Display is based on sample scores on the primary constrained axis (CAP1, 2.2% variance explained) and primary multidimensional scaling (MDS1, 20% variance explained). Lines connect samples taken at the same time point, and colored circles indicate the samples near the center of gravity for each time point. Below and left boxplots show the sample scores in CAP1 and MDS1 (boxes show medians/quartiles; error bars extend to the most extreme values within 1.5 interquartile ranges). Coding: 0, baseline; 1–3, during the swine farm stay; 4–6, 3 months after leaving the farm; 7–9, predicted time points over 3 months in the future ($n = 14$ biologically independent samples per timepoint for 0–6 and $n = 14$ predicted points per timepoint for 7–9). $*P < 0.05$; $**P < 0.01$; $***P < 0.001$; NS, not significant; Student's $t$-test (paired two-sided test between the students' samples at time point T0 versus time points T3, T6, T7, T8, and T9). Underlying data are provided in the Source Data file.

## Methods

**Ethics statement.** The Institutional Review Board of South China Agricultural University (SCAU-IRB) approved the protocols. All animals were sampled under authorization from Animal Research Committees of South China Agricultural University (SCAU-IACUC).

**Study design.** Fourteen senior class veterinary students (Student ID: H, I, J, K, L, M, N, O, P, Q, W, X, Y, Z) provided their written informed consent and voluntarily enrolled in the study during participation in an ~3-month-long practical training course in veterinary science at SCAU from July to October 2015. The 14 students were randomly divided into three groups of four to five persons, and each group was assigned to one of three swine farms in three different Chinese provinces, including (from north to south), Henan (Farm ID: H farm), Jiangxi (Farm ID: D farm), and Guangdong (Farm ID: S farm) (Supplementary Fig. 1a). These are typical large-scale swine farms, and all have been in operation for more than 5 years. Three farms implement self-breeding, and all use the closed-end management model. Among them, H farm is the largest, with 15,000 sows, D farm (7400 sows) is the next largest, and S farm (3800 sows) is the smallest. Due to limitations in the volunteer veterinary student population, all subjects were male and a parallel group of swine farm unexposed students was not possible. We have taken several steps to mitigate these limitations, including comparisons to a healthy cohort from urban Chinese individuals (BioProject accession number PRJEB13870 [https://www.ncbi.nlm.nih.gov/bioproject/PRJEB13870]). To control for differences at individual level, the students' fecal samples were collected longitudinally and the fecal samples before arriving at the farm (T0) were considered a blank control. In addition, four to five farm workers in each swine farm were also recruited in this study. All the farm workers had engaged in pig farming for 4–18 years and stayed at the present farm at least for 1 year. The volunteers signed an informed consent form and were asked to agree to fecal swabbing and to complete a short questionnaire related to personal information, such as age and gender, personal hygiene, dietary habits, antibiotic use, hospitalization, previous visits to farms or factories, and other pertinent factors (Supplementary Table 1). In addition to environmental exposure, other factors such as diet and work stress may be the important factors influencing the human gut microbiota. Considering that these factors may be caused by environmental changes, in this study, we consider these related factors as environmental impacts.

**Sample collection.** The students' fecal samples were collected at the following intervals: (1) 1–2 weeks prior to their entry into the swine farm, (2) weekly for the 3 consecutive months of their stay at the swine farm; (3) monthly for another 3 consecutive months after their return to the university. At each swine farm, four to five farm workers who had worked on the farm for at least one year were recruited, and their fecal samples were collected monthly during students' swine farm stays. In addition, averages of 40 pig feces samples, 3 soil samples, 3 sewage samples, and 3 ventilation dust samples for each farm, were collected monthly for the 3 consecutive months of the students stay at the swine farm. Among them, 42 students' fecal samples and 12 pooled samples consisting of 55 environmental samples (around 3–5 samples for each item per farm) from the swine farms, including pig feces, soil, sewage, and ventilation dust, were used in the metagenomic sequencing (Supplementary Data 3 and 5). Samples were submitted using an assigned student study ID and date. Samples were kept on dry ice during transport and were stored at −80 °C prior to DNA extraction and chemical analysis.

**DNA extraction.** Genomic DNA was extracted from all samples using the HiPure Stool DNA Kit (Magen, No. D3141) according to the manufacturer's instructions. Briefly, STL buffer (1 ml) was added to 50 mg of sample in a 2-ml screw-cap tube (Axygen), and the mixture was incubated at 65 °C for 10 min. The samples were then vortexed for 15 s and centrifuged at 13,000×$g$ for 10 min, and 600 μl of the supernatant was transferred to a fresh 2.0-ml tube. PS buffer (150 μl) and 150 μl of absorber solution was then added. Following a second centrifugation (13,000×$g$, 5 min), the supernatants were placed in fresh 2.0-ml tubes, and 700 μl of GDP buffer was added. A HiPure DNA Mini Column (Magen; No. D3141) was used to absorb the products, which were then eluted with sterile water.

**16S rRNA amplification, sequencing, and preprocessing.** The V3 and V4 hypervariable regions of the 16S rRNA gene were sequenced and analyzed to define the composition of the bacterial community in human fecal samples. The following amplification primers were used: primer-F = 5′ TCGTCGGCAGCGTCA GATGTGTATAAGAGACAGCCTACGGGNGGCWGCAG; primer-R = 5′ GTCTCGTGGGCTCGGAGATGTGTATAAGAGACAGGAC TACHVGGG TATCTAATCC. For amplicon library preparation, 20 ng of each genomic DNA, 1.25 U Taq DNA polymerase, 5 μl 10× Ex Taq buffer (Mg$^{2+}$ plus), 10 mM dNTPs (all reagents purchased from TaKaRa Biotechnology Co., Ltd), and 40 pmol of

primer mix was used for each 50-μl amplification reaction. For each sample, the 16S rRNA gene was amplified under the following conditions: initial denaturation at 94 °C for 3 min followed by 30 cycles of 94 °C for 45 s, 56 °C for 1 min, and 72 °C for 1 min and a final extension at 72 °C for 10 min. The PCR products were quantified by gel electrophoresis, pooled and purified for reactions. Pyrosequencing was performed on an Illumina MiSeq sequencer with paired-end reads 300 base pairs (bp) in length.

Based on the overlaps between the sequenced paired-end reads, the reads were merged into long sequences using the FLASH algorithm (min-overlap = 30, max-overlap = 150)[42]. Low-quality sequences were then trimmed and eliminated from the analysis based on the following criteria: (a) shorter than 400 bp and (b) a sequence producing more than 3 'N' bases. Bioinformatic analysis was implemented using the Quantitative Insights into Microbial Ecology QIIME2 platform (https://qiime2.org/)[43]. Briefly, raw Illumina amplicon sequence data was quality control processed using the DADA2 algorithm[44], removing the chimeric sequences and truncating the sequences from 5 to 250 bases. Phylogenetic diversity analyses were realized via the q2-phylogeny plugin, which used the mafft[45] program to perform multiple sequence alignment on the representative sequences (FeatureData in QIIME2) and the FastTree[46] program to generate phylogenetic tree from the alignments. The microbial community structure (i.e., species richness, evenness, and between-sample diversity) of fecal samples was estimated by biodiversity. The Shannon index was used to evaluate alpha diversity, and the weighted and unweighted UniFrac distances were used to evaluate beta diversity. All of these indices were calculated by the QIIME2 pipeline (q2-diversity plugin).

**Metagenomic sequencing and data quality control**. The Illumina HiSeq 3000 platform was used to sequence the samples. We constructed a 150-bp paired-end library with an insert size of 350 bp for every sample. The raw sequencing reads for each sample were independently processed for quality control using the FASTAX Toolkit (http://hannonlab.cshl.edu/fastx_toolkit/). The quality control used the following criteria: (1) reads were removed if they contained more than 3 'N' bases or more than 50 bases with low quality (<Q20); (2) no more than 10 bases with low quality (<Q20) or assigned as N in the tails of reads were trimmed. The remaining reads were then mapped to the human and swine genomes using SOAPalinger2[47] to remove host DNA contamination. Overall, an average of 0.9% of low-quality or host genome reads was removed from the sequenced samples.

**De novo assembly, gene calling, and gene catalog construction**. To determine the best assembling method for high-quality whole-metagenome-sequencing reads, we compared the performance of two assemblers, SOAPdenovo v2 (previously used in human gut microbiomes)[25,48] and MEGAHIT (a de novo assembler for large and complex metagenomic sequences)[49]. For SOAPdenovo, we tested the k-mer length ranging from 23 to 123 bp by 20-bp steps for each sample and selected the assembled contig set with the longest N50 length. For MEGAHIT parameters "–mink 21 –maxk 119 –step 10 –pre_correction" were used. For most of the samples, MEGAHIT obtained a better assembled contig set than SOAPdenovo; this could be due to its improved assembly of bacterial genomes with highly uneven sequencing depths in metagenomic samples. As a result, we obtained an average of 254.6 ± 72.4 and 754.4 ± 180.4 Mbp (mean ± SD) contig sets for human fecal samples and environmental samples, respectively. The unassembled reads for each ecosystem were pooled and reassembled for further analysis.

Genes were predicted by MetaGeneMark[50] based on parameter exploration by the MOCAT pipeline[26]. A non-redundant gene catalog was constructed using CD-HIT[51]; from this catalog, genes with >90% overlap and >95% nucleic acid similarity (no gap allowed) were removed as redundancies. The gene catalogs contained 3,338,109 and 11,374,480 non-redundant genes generated from the human microbiome and the swine farm ecosystem, respectively.

**Quantification of metagenomic genes**. The abundance of genes in the non-redundant gene catalogs was quantified as the relative abundance of reads. First, the high-quality reads from each sample were aligned against the gene catalog using SOAP 2.21[47] using a threshold that allowed at most two mismatches in the initial 32-bp seed sequence and 90% similarity over the whole read. Then, only two types of alignments were accepted: (1) those in which the entirety of a paired-end read could be mapped onto a gene with the correct insert size; (2) those in which one end of the paired-end read could be mapped onto the end of a gene only if the other end of the read mapped outside the genic region. The relative abundance of a given gene in a sample was finally estimated by dividing the number of reads that uniquely mapped to that gene by the length of the gene region and by the total number of reads from the sample that uniquely mapped to any gene in the catalog. The resulting set of gene relative abundances for all samples was termed a gene profile. The average read mapping rates (or mean reads usage) were 71.5% and 43.8% for human gut microbiome and swine farm environmental samples, respectively.

**Quantification of taxa in metagenomic data**. We performed the taxonomic profiling (including phylum, class, order, family, genus, and species levels) of the metagenomic samples using MetaPhlAn2[52], which relies on ~1 million clade-specific marker genes derived from 17,000 microbial genomes (including bacterial,

archaeal, and viral species) to unambiguously classify metagenomic reads to taxonomies and yield relative abundances of taxa identified in the sample.The Shannon index, was used to represent the within-sample diversity (alpha diversity) of the microbiota in the samples[53].

**Identification and quantification of AR genes**. The AR genes from each metagenomic assemblies were identified by blasting protein sequences against CARD (downloaded February 2018)[36] database using stringent cutoff (>95%ID and >95 overlap with subject sequence). The remaining unannotated sequences were filtered and subsequently annotated with Resfams core database. This approach resulted in 12,739 unique AR genes from 66 metagenomic assemblies. Together, these 12,739 genes with 2252 AR sequences from CARD database were used to create high-precision sequence markers using ShortBRED[35] (parameters: –clustid 0.95 and –ref Uniref90.fasta).

The ShortBRED results included 20,514 markers for 5607 AR gene families. The marker list was then manually curated to reduce the rate of false positives in our surveys. Following criteria was used to filter out the false positives:

- genes that confer resistance via overexpression of resistant target alleles (e.g. resistance to antifolate drugs via mutated DHPS and DHFR);
- global gene regulators, two-component system proteins, and signaling mediators;
- efflux pumps that confer resistance to multiple antibiotics;
- genes modifying cell wall charge (e.g. those conferring resistance to polymixins and defensins).

The final set consisted of 1924 AR gene families. The abundance of AR gene families was measured using shortbred_quantify.py script and about 1018 AR determinants were detected with RPKM > 0 in at least two samples.

**Identification of virulence factor genes and antibacterial biocide and metal resistance genes**. We identified the virulence factors based on the Virulence Factors of Pathogenic Bacteria Database (VFDB, downloaded February 2018)[54] and the antibacterial biocide and metal resistance genes based on the BacMet database[55]. Amino acid sequences were aligned against the databases using BLASTP (e-value ≤ 1e−5) and assigned to genes by the highest-scoring annotated hit with >80% similarity that covered >70% of the length of the query protein.

**Species transmission event identification and SourceTracker**. We used a modified SourceTracker algorithm[37] to identify species transmission events from the swine farm environment to human gut microbiota. Briefly, the new genes found in each sample during swine farm residence were grouped into species-level clusters by consistent taxonomic assignment and relative abundance (range: average ±5%). The SourceTracker algorithm was then used to estimate the probability that the species in the fecal sample came from the source environment (probability > 80%). The probable transferred species with <100 genes or <0.01% relative abundance in the human gut microflora were further filtered.

To identify transfer events involving AR genes, SourceTracker was run with the default settings using the environmental microbiota as the source.

**Microbial genome reconstruction in metagenomes**. We established an approach to reconstruct the genomes of the high-abundance (typically, >3%) species in the human gut metagenomes. Firstly, metagenomic reads were mapped to the closest reference genomes using SOAP2.21[47] (>95% identity). The mapped reads were independently assembled using Velvet[56], an algorithm for de novo short read assembly for single microbial genomes. The software was run multiple times using different k-mer parameters ranging from 39 to 131 to generate the best assembly results. Then, the raw assembled genome was scaffolded by SSPACE[57], and gaps were closed by GapFiller[58]. The short scaffolds were filtered with a minimum length threshold of 200 bp. A circle plot of the draft genomes was obtained using BRIG software[59]. The average nucleotide identity (ANI) between genomes was calculated using the ANIb algorithm, which uses BLAST as the underlying alignment method[60].

**Network visualization**. The AR gene co-occurrence network was visualized by Cytoscape 3.3.0[61] using an edge-weighted spring-embedded layout.

**Mobile genetic elements**. Putative MGE genes, including transposase, integrase, recombinase, phage terminase and endopeptidase genes, and bacterial insertion (IS) sequences were identified from the functional selection by Pfam (v29.0)[62] and Kyoto Encyclopedia of Genes and Genomes (KEGG, downloaded December 2017)[63] annotation. AR genes were considered to co-localize with an MGE if they shared a contig with an MGE gene in a nearby area (<10 kilobases).

**Phylogenetic classification of contigs**. AR contigs and metagenomic assembly contigs were classified using BLASTN with parameters "-word_size 16 -evalue 1e−5 -max_target_seqs 5000" based on the NCBI reference microbial genomes (downloaded December 2017). At least 70% alignment coverage of each contig reads was required. Based on the parameter exploration of sequence similarity

across phylogenetic ranks[64], we used 90% identity as the threshold for species assignment and 85% identity as the threshold for genus assignment.

**Cultures and *E. coli* analyses**. All samples were cultured on MacConkey agar plates and incubated at 37 °C for 24 h. Five to six suspicious colonies with typical *E. coli* morphology was selected from each sample for identification. We obtained 1851 *E. coli* isolates, including 954 isolates from students' fecal samples, 182 isolates from farm workers' fecal samples, 657 isolates from pig feces, and 58 isolates from other farm environmental samples (soil, sewage, and ventilation dust). After identifying *E. coli* isolates by MALDI-TOF MS (Biomerieux, France), we characterized 82 *E. coli* isolates to determine their genetic relatedness by pulsed-field gel electrophoresis (PFGE)[65]. These 82 *E. coli* strains were randomly selected from one pig farm and origin from students ($n = 13$), farm workers ($n = 2$), pigs ($n = 51$), and farm environments (16). The DNA banding patterns were analyzed by Bio-Numerics software (Applied Maths, Sint-Martens-Latem, Belgium) using the Dice similarity coefficient and a cut-off value of 85% of the similarity values was chosen to indicate identical Eric types. *Salmonella enterica* serotype Braenderup H9812 standards served as size markers.

**Phenotypic and genotypic resistance testing**. All 1851 *E. coli* isolates were tested for susceptibility to 11 antimicrobials for human medicine and food animals' production, including colistin (CS), cefotaxime (CTX), gentamicin (GEN), amikacin (AMK), tetracycline (TET), fosfomycin (FOS), ciprofloxacin (CIP), methoxazole/trimethoprim (S/T), chloromycetin (CHL), meropenem (MEM), and tigecycline (TIG). Antimicrobial susceptibilities of isolates were determined by the agar dilution method and the results were interpreted according to the Clinical and Laboratory Standards Institute (CLSI) (M100-S25)[66]. All isolates were further screened for $bla_{\text{CTX-M}}$ and $fosA3$ genes (conferring resistance to CTX and FOS, respectively) by PCR amplification using primers (Supplementary Table 2). As $fosA3$ was frequently co-transferred with $bla_{\text{CTX-M}}$ mediated by a single plasmid[24], the genetic contexts of the $fosA3$ and $bla_{\text{CTX-M}}$ genes were explored by PCR mapping (Supplementary Table 2) using the reference regions surrounding them among 15 $fosA3$-$bla_{\text{CTX-M}}$-co-harboring *E. coli* isolates, which were randomly selected from one pig farm.

**Creation of the DBN model**. The DBN model was created based on genus composition profiles of students' fecal samples at all seven time points. Firstly, we removed (1) two students (H and N) who lacked the sequencing data for at least two time points, and (2) the genera with average relative abundance <0.5% in students, remaining the gut microbial communities of 12 students on 39 high-abundant genera for further analysis. These genera covered 86% of total relative abundance of analyzed samples. Then, we calculated the genus–genus associations based on the extended local similarity analysis (eLSA) algorithm[67] (default parameters), using the students' genus profiles at all seven time points. The eLSA tool generated an association network from significant associations (permutated $P <$ 0.01), including both time-independent (undirected) and time-dependent (directed) associations. For each genus, five most significant associations were remained to simplify the network. Lastly, the partially directed DBN model was created based on the genus-genus association network and the directed associations for each genus from its previous time point to current time point (as shown in Supplementary Fig. 15).

**Prediction of the microbial composition based on the DBN model**. In the DBN model, the current relative abundance ($t_n$) of every genus can be expressed as a function of the relative abundances of its parent genera at the previous time point ($t_{n-1}$). The functions in the resulting DBN were derived using Eureqa v1.24.0[6] (default parameters). Eureqa is a freely downloadable software for deducing equations and hidden mathematical relationships in numerical data sets without prior knowledge of existing patterns. The operations, including constant, add, subtract, multiply, divide, sine, cosine, and exponential, were permitted in solutions. Eureqa was allowed to search for best-fitting equations for a maximum of $1 \times 10^{10}$ formula evaluations, or until correlations >0.8 were observed. To evaluate the accuracy of the DBN model, we trained a new model by using the microbial compositions at time points T0–T5 and then predicted the microbial composition at T6. This leave-one-out cross-validation strategy was also used to predict the compositions of time points T1–T5. For all samples, their predicted microbial compositions achieved high consistency by Bray–Curtis similarity (1-Bray–Curtis distance). Finally, in our dataset, we predicted the relative abundance of all genera at an extrapolated time point (T7) based on the formulas, using their abundances at time point T6. Similarly, the microbial communities at time points T8 and T9 were predicted based on T7 and T8.

**Statistical analysis**. Statistical analysis was implemented using the R platform. Principal coordinate analysis (PCoA) was performed using the "ape" package[68] based on the UniFrac distances between samples. dbRDA was performed using the "vegan" package[69] based on the Bray–Curtis distances on normalized taxa abundance matrices and visualized using the "ggplot2" package. In analyses of PCoA and dbRDA, the top two principal components of the samples were shown, and the Mann–Whitney $U$-test was used to evaluate the significance of differences in

samples obtained at different time points. PERMANOVA[70] was used to determine the significance of time points on the subject's gut microbiota as well as antibiotic resistome. We implemented PERMANOVA using the adonis function based on the Bray–Curtis dissimilarity and 999 permutations. This function calculates the interpoint dissimilarities of each group and compares these values to the interpoint dissimilarities of all points to generate a pseudo-$F$ statistic. This pseudo-$F$ statistic is then compared to the distribution of pseudo-$F$ statistics generated when the function is run on the dissimilarity matrix with permuted labels. Procrustes analysis was performed using the "vegan" package, and the significance of the Procrustes statistic (a correlation-like statistic derived from the symmetric Procrustes sum of squares) was estimated by the protest function with 999 permutations. Rarefaction analysis implemented by in-house Perl scripts was performed to assess the gene richness of environmental samples. Statistical significance was set at $P < 0.05$ following Benjamini–Hochberg corrections.

**Reporting summary**. Further information on research design is available in the Nature Research Reporting Summary linked to this article.

## Data availability
Sequence data that support the findings of this study are available in the European Nucleotide Archive under BioProject number PRJEB20626 (https://www.ebi.ac.uk/ena/data/view/PRJEB20626). The source data underlying Figs. 1b, c, 2a–c, 3a, b, 4a, 4c, d, and 5 and Supplementary Figs. 2, 3a, b, 4a–e, 5a, b, 7a–d, 8, 9, 11a–d, 12a, b, 13, 14, and 15c are provided as a Source Data file.

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

## Acknowledgements

We thank the 14 senior class veterinary students from South China Agricultural University, and the farmer managers and workers from our study typical large-scale swine farms in three different Chinese provinces, without which this study would not have been possible. This work was jointly supported by the National Key Research and Development Program of China (2016YFD0501300 to Y.-H.L.), the Program for Changjiang Scholars and Innovative Research Team in University of Ministry of Education of China (IRT_17R39 to Y.-H.L.), the Foundation for Innovation and Strengthening School Project of Guangdong (2016KCXTD010 to Y.-H.L.), the National Natural Science Foundation of China (31730097 to Y.-H.L.), the 111 Project (D20008 to J.S., X.-P.L., and Y.-H.L.), the Institutional Program Unifying Population and Laboratory-Based Sciences Burroughs Wellcome Fund grant to Washington University (supporting A.W.D.), and the National Institutes of Health (NIH) Director's New Innovator Award (to G.D.). The content is solely the responsibility of the authors and does not necessarily represent the official views of the funding agencies.

## Author contributions

Y.-H.L., G.D., J.S., and X.-P.L. designed this study; K.C., L.L., T.H., J.X., Y.Y., Y.-F.Z., Y.-X.S., and X.-B.D. collected surveys and samples from student volunteers and swine farms; K.C., L.L., T.H., Z.-W.F., and Q.-L.Y. extracted DNA and generated 16S, functional metagenomic, and shotgun data for samples from student volunteers and swine farms; L.-X.F., X.-L.L., and R.-M.Z. performed cultures and *E. coli* analyses, and phenotypic and genotypic testing; J.S., X.-P.L., A.W.D., M.B., S.-H.L., J.L.M., Y.-J.F., Z.-L.Z., H.-X.J., B.-H.F., Y.-Z.T., G.D., and Y.-H.L. performed analyses and interpreted results; J.S., X.-P.L., A.W.D., M.B., S.-H.L., J.L.M., G.D., and Y.-H.L. wrote the paper with input from other co-authors.

## Competing interests

The authors declare no competing interests.
