## [Peer Review File · Nature Communications]

Reviewers' comments:

Reviewer #1 (Remarks to the Author):

In this paper, the authors focused on studying the impact of swim farm environments on changes in gut microbiome and resistomes of veterinary students. They identified antibiotic resistance genes in similar genetic contexts in student samples and farm environmental samples. The writing needs to be improved, especially some important details are missing in the description of the statistical methods. My comments will be focused on the study design and data analysis methods.

1. It would be clearer if the authors provide a detailed introduction for the entire dataset that were used for the analysis. It is unclear to me which information is available and which information is unavailable and needs us for the statistical inference?

2. It is not clear to me if the authors would like to draw conclusion for single-subject analysis or for multiple-subject analysis. If the conclusion is interpreted at a population level, the 14 healthy male subjects seem very limited. How much representative these 14 males are? How about female subjects?

3. On Page 3, in "Gut microbiota changes correlate with environmental exposure." It would be helpful to provide more details in multivariate analysis of operational taxonomic unit (OTU) composition? What is PERMANOVA? It would be clearer if the authors provide references or briefly introduce the model details.

4. This study did not have the parallel control cohort. Instead, the students' fecal samples were collected longitudinally and the fecal samples before exposing at the farm were considered a black control. The statistical analysis results may be affected by time-dependent confounding factors. Did the authors adjust those confounding factors in the analysis?

5. In Figure 1b, the first two constrained principal coordinates only explain 1.8% and 1.3% of variance. Therefore, the majority of variance is explained by rest of coordinates. How consistency of the sample paths on the first two coordinators with those on the rest coordinates.

6. It is unclear to me how the authors perform the extended local similarity analysis to model the dynamic Bayesian network. What are the prior specifications for the model? What prior knowledge did the authors include in the model? How sensitive are the posterior results to the prior specifications? In addition, Figure 5 is less informative to show the prediction power. It would be most helpful if the authors report the predictive R-squared and/or predictive mean squared errors.

Reviewer #2 (Remarks to the Author):

The manuscript entitled "Environmental remodeling of human gut microbiota and antibiotic resistome in livestock farms" describes the spread and increased abundance of antibiotic resistance bacteria and genes from environmental sources to the guts of humans on residing on swine farms. The data described that overall composition and types of ARGs of the student's microbiota became more similar to that of farm workers after 3 months but then declined after 3 months of leaving the farms. The data suggest this is a generalizable phenomenon as the acquisition of ARB/ARGs was similar at three geographically separated farms. I found the manuscript to be well written and the complex data very nicely presented. I also liked the fact the authors used both NGS and culturing methods.

I have one major sticking point around the notion of detecting an HRG event using the methods herein. For instance, on lines 162-170 (and other places) the authors suggest that they were able to track the flow of ARGs from farm sources to the guts of students. However, these observations are based on microbiota level data or by assembly of metagenomic data to provide context within the genetic elements that should promote ARG transfer. This approach is great but these analyses can't discriminate between ARGs of organisms acquired by students from the environment versus the transmission of ARGs from the environment into bacteria already present in the guts of the students. I feel the E. coli PFGE data supports this and that after 3 months leaving the farm, the student's gut microbiota was more similar to a pre-farm exposure status. In short what has been described is the acquisition of ARGs via farm derived ARBs by students, not the transfer of ARGs from the farm environment or ARBs into existing gut microbiota. Thus, the language surrounding gene transfer should be modified to reflect this. Otherwise the manuscript is great.

Reviewer #3 (Remarks to the Author):

I have with interest read the study that further supports that it is important to consider also transmission and not only selection when dealing with antimicrobial resistance.

Even though it is a very well known occupational risk that microbes and AMR genes will transmit from the environment to those working there, the research question regarding how fast and whether the microbiome reverts is certainly worth studying.

However, while certainly of scientific merit I think the authors are not entirely capturing this and that they are drawing their conclusions a bit too far. Especially regarding the metagenomic part of the study.

Thus, as I interpret figure 1 and figure 5, there is a drastic, but systematic change in the microbiome during the first 3 months, the composition is then jumping back and forth for the next 3 months, but convenient with month 6 closest to the starting point. None-the-less the prediction suggest that the composition will never return to the starting point. I know that the authors conclude “partially reverse”, but it would have been nice with at least two time-points in the end reasonably close together.

The resistome analyses are based on only three time-points and naturally since time point 6 is closest to the start it will suggest that the resistome reverts. It is a bit disturbing that the abundance median for time 6 is the highest. The *E. coli* data do however, very nicely support that this is the case and consequently makes this point stronger.

Other comments:

In the introduction

- I am not sure reference 20 proves that antibiotics are used indiscriminately and this is a hot-spot for enrichment and exchange. As far as I recall reference 20 is looking at the microbial composition and resistome in sewage in a few selected sites, but with no AMU data and no study regarding tracing back AMR genes to humans

- Since growth promoters are not used in all countries the authors should perhaps mention that this is for China

- Mcr-1 was first reported from China, but have been found in older samples elsewhere
- Perhaps the authors should refer to some studies which already have shown environmental transmission. To my knowledge the first showing this for livestock was Levy et al. Nature 260, pages 40–42 (1976).

Gut changes

I think it is appropriate if the authors acknowledge more that there can be other reasons for change. Thus, if you as these students probably do go from an urban environment to a rural, other systematic changes such as diet. I know they mention it, but seem to ignore it again.

Microbial source tracking

While certainly relevant, all these source attribution models whether random forest or the source tracker depends on information regarding as many reservoirs as possible. I am actually a bit surprised that the workers constitute a source of the same since as the pig feces. This surely suggests that other sources should be taken into account.

E. coli data

This looks very convincing, but is the prevalence of workers and environment really that stable?

Response to referees for manuscript **NCOMMS-19-13605A**, “**Environmental remodeling of human gut microbiota and antibiotic resistome in livestock farms.**” by Jian Sun, Xiao-Ping Liao, Alaric W. D'Souza*, Manish Boolchandani, Sheng-Hui Li, Ke Cheng, José Luis Martínez, Liang Li, You-Jun Feng, Liang-Xing Fang, Ting Huang, Jing Xia, Yang Yu, Yu-Feng Zhou, Yong-Xue Sun, Xian-Bo Deng, Zhen-Ling Zeng, Hong-Xia Jiang, Bing-Hu Fang, You-Zhi Tang, Xin-Lei Lian, Rong-Min Zhang, Zhi-Wei Fang, Qiu-Long Yan, Gautam Dantas, and Ya-Hong Liu.

We thank the referees and others involved in the editorial process, for their time and effort in considering this manuscript and their thoughtful suggestions to help improve it. We have completed additional analysis and made figures to address their concerns. We have also included additional information, textual edits, and analysis in our manuscript thanks to the reviewer remarks. Having addressed all reviewer concerns, our revised manuscript is greatly strengthened, and we hope we are a stronger candidate for publication in *Nature Communications*.

We respond to individual comments from referees below (reviewer comments are on the left and author responses are on the right):

Responses to Reviewer 1.	
In this paper, the authors focused on studying the impact of swim farm environments on changes in gut microbiome and resistomes of veterinary students. They identified antibiotic resistance genes in similar genetic contexts in student samples and farm environmental samples. The writing needs to be improved, especially some important details are missing in the description of the statistical methods. My comments will be focused on the study design and data analysis methods.	We thank the reviewer for summarizing our manuscript and for presenting us with organized revision recommendations. Below we detail our attempts to revise our manuscript in accordance with these suggestions.
It would be clearer if the authors provide a detailed introduction for the entire dataset that were used for the analysis. It is unclear to me which information is available and which information is unavailable and needs us for the statistical inference?	In our revised manuscript, we replaced Figure 1a with a new flowchart to illustrate our study design and datasets. Following is the description of the datasets using in this study. We longitudinally collected fecal samples from 14 students at 7 time points, including upon arrival at the swine farm (T0), during the swine farm stay (T1-T3, monthly), and after leaving the farm (T4-T6, monthly). First, we analyzed all samples via 16S rRNA gene sequencing to investigate the overall alteration of the participants' gut microbiota, which correspond to the first section of results (“Gut microbiota changes correlate with environmental exposure”). Then, to reify alterations to the gut microbiome and investigate potential changes in the antibiotic resistome, we whole-metagenome shotgun sequenced the participants' fecal samples at T0, T3 and T6, which correspond to the second section of results (“Antibiotic resistome structure influenced by changing environment”). We also whole-metagenome shotgun sequenced four types of

	environmental samples of the swine farms, to explore the potential exchange of microbes and antibiotic genes between the swine farm environment and the students' gut, which correspond to the third and fourth sections of results (“Microbial transmission from the swine farm environment into students' gut”, and “Transfer of antibiotic resistance genes between the swine farm environment and human gut”). Lastly, we used a time series model to predict the persistent period of impact from the swine farm to the students' gut, which correspond to the fifth section of results (“Predicting the impact of environmentally induced changes on the gut microbial community structure”).
It is not clear to me if the authors would like to draw conclusion for single-subject analysis or for multiple-subject analysis. If the conclusion is interpreted at a population level, the 14 healthy male subjects seem very limited. How much representative these 14 males are? How about female subjects?	The main conclusions of our study were based on multiple-subject analysis. We agree with the reviewer about our cohorts limitations at the population level, and have declared these limitations in the revised manuscript based on the reviewer's suggestion (Lines 371-373). The primary purpose of our study was to investigate the potential impact of swine farm environment on the participants' gut microbiota and antibiotic resistome. Based on our analyses, we report significant observed alterations of participants' gut microbiome/resistome during and after their swine farm stay, and also identified some microbial and antibiotic resistance gene transmission events from the farm environment into human gut. Particularly, these findings were observed in almost all participant students (see Figure 1b and Figure 3b in the manuscript). Therefore, individual differences of students have little influence on the final conclusion of the manuscript. Considering that the difference between the gut microbiota of male and female is relatively limited¹, we believe that the conclusion of our manuscript would repeat in female subjects; however, we do directly acknowledge this biological sex caveat in our manuscript on Line 281 and Lines 371-373.
On Page 3, in “Gut microbiota changes correlate with environmental exposure.” It would be helpful to provide more details in multivariate analysis of operational taxonomic unit (OTU) composition? What is PERMANOVA? It would be clearer if the authors provide references or briefly introduce the model details.	Thanks for your suggestion. In Figure 1b, we performed a distance-based redundancy analysis (dbRDA) of Bray-Curtis distances between operational taxonomic units (OTUs) in samples at all-time points. dbRDA is shown at the first two constrained principal coordinates, which revealed the gut microbiota was altered during the students' swine farm stays and partially recovered after leaving the farm. The time point- derived alteration of students' gut microbiota was significant, as revealed by the effect size ($R^2 = 7.4\%$) and permutation test

	($P < 0.001$) via PERMANOVA analysis. We modified these descriptions in the revised manuscript to clarify the results and methods, as the reviewer suggested. Lines 620-625. Permutational multivariate analysis of variance² (PERMANOVA) is a relatively general method that used in bacterial community analysis to illustrate the influence of bacterial community composition of single or multiple confounding factors³. It quantifies the variance of each confounding factor based on a distance matrix and uses a permutation test to estimate the significance of the factor. This test differs from a normal MANOVA because the input matrix is a semi-metric dissimilarity matrix as opposed to a metric distance matrix. Rather than calculating an F-statistic using the centroids for the two groups compared to the centroid for the center, the algorithm instead calculates the interpoint distances. This is because with semi-metric distances, the triangle inequality may be violated (centroids are not easily calculated), so the pairwise distances between samples must instead be used. The pseudo-F statistic generated by these interpoint distances (within group interpoint distances vs all samples interpoint distances) is then compared to permutation calculations for the pseudo-F statistic to get the final p-value. Based on the reviewer's suggestion, we added a brief introduction and references for PERMANOVA in the Method section of the revised manuscript. Please see Lines 96 and 620-625.
This study did not have the parallel control cohort. Instead, the students' fecal samples were collected longitudinally and the fecal samples before exposing at the farm were considered a black control. The statistical analysis results may be affected by time-dependent confounding factors. Did the authors adjust those confounding factors in the analysis?	We agree with the reviewer that time-dependent confounding factors, e.g. seasonal change, could impact the participants' gut microbiota. Previous studies⁴ revealed that, during seasonal variation, the gut microbiota of contemporary human populations (e.g. westernized Americans) was relatively stable, with some shifts in the abundance of particular taxa. To reduce potential effects from time-dependent confounding factors, we compared the gut microbiomes of our participants with a baseline healthy cohort of 196 urban Chinese subjects (their fecal samples were collected in all seasons)⁵, and found that the students' gut microbiota at T3 was significantly more dissimilar from the baseline samples than T0 or T6 times (Extended Data Figure 4d), suggesting our findings are likely to be robust across different populations. These descriptions were clarified in the revised manuscript based on the reviewer's comment. Please see Lines 122-125.

In Figure 1b, the first two constrained principal coordinates only explain 1.8% and 1.3% of variance. Therefore, the majority of variance is explained by rest of coordinates. How consistency of the sample paths on the first two coordinators with those on the rest coordinates.	As dbRDA analysis was constrained by the variables, the explanation rate of the principal coordinates depended on the overall effect size of the variables. In our study, based on PERMANOVA analysis, we found that the time point variable explained a total of 7.4% of gut microbiome variation. This effect size was small but significant ($P < 0.0001$ under 10000 permutations). Due to the complexity and mutability of gut microbiota, measured variables often have this level of impact on microbial community composition. For example, a recent study¹ quantified the effect size of 503 clinical factors, including blood parameters, medication, diet, and lifestyle, on gut microbiota, and identified 69 factors with significant impact. The maximum explanation rate of these factors was 5% (highest, 4.8%), and a combination of these factors showed an effect size of 16.4% on gut microbiome variation. Considering these published results, the impact of swine farm time point on participants' gut microbiota was considerable. To investigate consistency of the sample paths in the other coordinates, we made additional figures with all pairwise comparisons of the first 4 coordinates of our dbRDA analysis (shown below). The trends for the analysis are similar across the plots, especially with respect to time point 0 and time point 6: It is unclear to me how the authors perform the extended local similarity analysis to model the dynamic Bayesian network. What are the prior specifications for the model? What prior knowledge did the authors include in the model? How sensitive the posterior results to the prior specifications? In addition, Figure 5 is less informative to show the prediction power. It would be most helpful if the authors report the predictive R-squared and/or predictive mean squared errors.	Usually, the microbes in a normal gut microbial community are tightly correlated, and the data is inherently compositional. Thus, the abundance of a microbe at a given time point may be predicted by the abundances of itself and all other microbes at the previous time points. This hypothesis is limited since it does not account for environmental factors, but to a large extent, it can effectively capture the dynamic variation of microbial community over time (see below).

	In this study, we first calculated the correlation of 39 high-abundance genera using the extended local similarity analysis (eLSA) algorithm based on their relative abundances at 7 time points (T0-T6). The eLSA generated a genus-genus association network of significant correlations ($P < 0.01$), including both time-independent (undirected) and time-dependent (directed) correlations. For each genus, the five most significant associations were retained to simplify the network. Then, a dynamic Bayesian network (DBN), showing at Extended Data Fig. 15) was created by combined the partially directed genus-genus association network and the directed associations for each genus from its previous time point to current time point. In the DBN model, the current relative abundance (t_n) of every genus can be expressed as a function of the relative abundances of its parent genera at the previous time point (t_{n-1}). The functions in the resulting DBN were derived using Eureqa, a software to deduce the best-fitting equations and hidden mathematical relationships without prior knowledge of existing patterns. Finally, using the DBN model and formulas, we predicted the relative abundance of all genera at an extrapolated time point (T7) based on their abundances at time point T6. Similarly, the microbial communities at time points T8 and T9 were predicted based on T7 and T8 respectively. We have added additional text in the methods section Lines 605-610 to help clarify this method. We have also added additional figures and text to Extended Data Figure 15 to depict our model validation strategy. Lines 921-924. To evaluate the accuracy of the DBN model, we trained a new model by using the microbial compositions at time points T0-T5 and then predicted the microbial composition at T6. This leave-one-out cross-validation strategy was also used to predict the compositions of time points T1-T5. For all samples, their predicted microbial compositions achieved high consistency by Bray-Curtis similarity (1-Bray-Curtis distance). See following figure, also shown in Supplementary Figure 15b, comparing predictions to measured microbial profiles, suggesting a high accuracy of our models.
--	--

Responses to Reviewer 2

The manuscript entitled “Environmental remodeling of human gut microbiota and antibiotic resistance in livestock farms” describes the spread and increased abundance of antibiotic resistance bacteria and genes from environmental sources to the guts of humans on residing on swine farms. The data described that overall composition and types of ARGs of the student’s microbiota became more similar to that of farm workers after 3 months but then declined after 3 months of leaving the farms. The data suggest this is a generalizable phenomenon as the acquisition of ARB/ARGs was similar at three geographically separated farms. I found the manuscript to be well written and the complex data very nicely presented. I also liked the fact the authors used both NGS and culturing methods.

We thank the reviewer for summarizing our work and for their kind comments regarding our manuscript. In particular, we greatly appreciate that they recognized the merit of both using NGS and culturing methods presented in this manuscript.

I have one major sticking point around the notion of detecting an HRG event using the methods herein. For instance, on lines 162-170 (and other places) the authors suggest that they were able to track the flow of ARGs from farm sources to the guts of students. However, these observations are based on microbiota level data or by assembly of metagenomic data to provide context within the genetic elements that should promote ARG transfer. This approach is great but these analyses can’t discriminate between ARGs of organisms acquired by students from the environment versus the transmission of ARGs from the environment into bacteria already present in the guts of the students. I feel the E. coli PFGE data supports this and that after 3 months leaving the farm, the students gut microbiota was more similar to a pre-farm exposure status. In short what has been described is the acquisition of ARGs via farm derived ARBs by students, not the transfer of ARGs from the farm environment or ARBs into existing gut

We agree with Reviewer 2’s concern about ambiguity surrounding whether ARG acquisition is caused by the transfer of ARBs or ARGs between the swine farm environment and humans.

To help illuminate this concern to potential readers, we have added several lines of text to make clear that lateral transfer of ARGs and transmission of ARBs are both plausible explanations for our observed data and analysis.

Lines 68-79

“Direct evidence has shown environmental transmission of AR genes and their bacterial hosts among livestock and humans.”

Lines 74-76

“This connectivity includes both the transmissibility of antibiotic resistant bacteria selected in animals to human hosts as well as the potential of lateral AR gene transfer between animal associated and human associated bacteria.”

microbiota. Thus, the language surrounding gene transfer should be modified to reflect this. Otherwise the manuscript is great.	Lines 169-173 “Since genes can transfer via host bacterial transmission or lateral gene exchange, we looked to correlate our gene exchange results with the OTU composition change of the students during their swine farm stays. Strong association between the OTU composition changes and gene exchange would suggest extensive microbial transmission between the environment and the human gut.” We have also modified text in the section where we make the case for the likely scenario being transfer of ARBs. Combining metagenomics and culturing methods, we not only found nearly two-thirds of new genes in student microbiota during swine farm residence were present in the environmental microorganism (Fig. 3a; Extended Data Fig. 8), but also a substantial number of putative transmission events (more than 142) were identified from various swine farm environments to the students’ gut microbiota (Fig. 3b), which was validated by SourceTracker and traditional PFGE. These confirm that the acquisition of ARGs is closely related to the transmission events of AGBs. Accordingly, per the Reviewer 2’s suggestion, we have rephrased the appropriate description. Please see sections titled “Microbial transmission from the swine farm environment into students’ gut” and “Transfer of antibiotic resistance genes between the swine farm environment and human gut”.
Responses to Reviewer 3	
I have with interest read the study that further supports that it is important to consider also transmission and not only selection when dealing with antimicrobial resistance. Even though it is a very well known occupational risk that microbes and AMR gens will transmit from the environment to those working there, the research question regarding how fast and whether the microbiome reverts is certainly worth studying. However, while certainly of scientific merit I think the authors are not entirely capturing this and that they are drawing their conclusions a bit too far. Especially regarding the metagenomic part of the study.	We are happy that the reviewer found our work of interest and we thank the reviewer for their helpful comments for revisions. Below we document our changes to the manuscript in response to these suggestions.
Thus, as I interpret figure 1 and figure 5, there	We designed our initial study to have 7 time points

is a drastic, but systematic change in the microbiome during the first 3 months, the compositing is then jumping back and forth for the next 3 months, but convenient with month 6 closest to the starting point. None-the-less the prediction suggest that the composition will never return to the starting point. I know that the authors conclude “partially reverse”, but it would have been nice with at least two time-points in the end reasonably close together.	based on our previous estimates of recovery time. The study was also partially limited by inability to collect samples during the Chinese Spring Festival where most students leave school and return home. The calculation and the time limitation resulted in us having 3 post-swine farm collection times spaced 1 month apart. As the reviewer mentions, in our sequencing analysis, we found at the T6 time point that the student's gut microbiome taxonomic composition partially recovered to the level before entering the farm, but resistome abundance was still maintained at a high level. The E. coli data also supports this point. To further explore the possible recovery time, we built Dynamic Bayesian Network models based on the data from the seven time points. With this model, we can predict a future time point based on the data of the previous time points. However, with this model deviations are amplified the further out in time one projects. Thus, if there is partial deviation of the previous point, subsequent time points will deviate further. This helps explain why our last two points are not perfectly overlapping with the T0 starting point. Additionally, the human microbiome is feature rich and dynamic, thus even repeated measures of the same individual within a span of days may have some natural variation when modeled. To clarify this point for the reader we have added the following text in the section discussing figure 5. “The change in taxonomic community structure stalls and stabilizes in the 6-9 month range. The time point at 9 months begins to have a wider distribution as it is further removed from the last measured sample.” See Lines 265-267.
The resistome analyses are based on only three time-points and naturally since time point 6 is closest to the start it will suggest that the resistome reverts. It is a bit disturbing that the abundance median for time 6 is the highest. The E. coli data do however, very nicely support that this is the case and consequently makes this point stronger.	The reviewers point here is well taken that the T6 time point has a high median of resistance gene abundance. Owing to our plan to perform 16S rRNA gene sequencing and culturing methods, we only selected T0, T3, and T6 to perform metagenomic (resistome) analyses, so we cannot directly track the trends between T3 and T6 for these resistance genes. It is possible that bacteria harboring resistance genes expanded within the microbiome community over time and were in a dynamic upward trajectory during the swine farm stay. If so, the results could be explained by the continuation of that trajectory following the students leaving the swine farms. As the reviewer notes, this was part of the motivation behind the culturing component of our manuscript.

	Similar to the trends of shift in the students' gut microbiota composition, the resistome showed marked deviation at T3 from the pre-exposure time-point (T0). But, the resistome didn't return to the starting point level at T6, though the student's gut microbiome had partially recovered. We observed an increase in the relative abundance of Proteobacteria (especially Gammaproteobacteria), during the study participants over the period from swine farm arrival (T0) to leaving the farm environment (T3). Thus, we isolated 1851 E. coli strains isolated from students, farm workers, and environment samples. Phenotypic resistance testing showed the similar result, the resistance rates maintained a high level once the students entered the farm. These descriptions were clarified in the revised manuscript based on the reviewer's comment (Lines 242-245).
I am not sure reference 20 proves that antibiotic are used indiscriminately and this is a hot-spot for enrichment and exchange. As far as I recall reference 20 is looking at the microbial composition and resistome in sewage in a few selected sites, but with no AMU data and no study regarding tracing back AMR gens to humans	Thank you for looking into this reference in more detail. We have clarified the sentence surrounding this reference to more accurately reflect the findings from the manuscript. "A recent study demonstrated that environments harbor microbial communities which can serve as hot-spots of resistance gene enrichment and exchange." Please see Lines 62-63.
Since growth promoters are not used in all contries the authors should perhaps mention that this is for China	We have modified the introduction section of the manuscript to specify that we are referring to China. Please see Line 65.
Mcr-1 was first reported from China, but have been found in older samples elsewhere	Thank you for catching this error. We have changed "discovered" to "reported". Please see Line 67.
Perhaps the authors should refer to some studies which already have shown environmental transmission. To my knowledge the first showing this for livestock was Levy et al. Nature 260, pages40–42 (1976).	As the reviewer suggested, we have added this reference. Please see Lines 68-69.
Gut changes I think it is appropriate if the authors acknowledge more that there can be other reasons for change. Thus, if you as these students probably do go from an urban environment to a rural, other systematic changes such as diet. I know they mention it, but seeming ignore it again.	We agree that many other factors may contribute to these differences, including but not limited to age, gender, personal hygiene, dietary habits, antibiotic use, and stress. We clarify this in our revision in the section "Gut microbiota changes correlate with environmental exposure", where we state "Since many

	environmental factors, including diet, antibiotics, and geography, have been associated with changes to the human gut microbiota, it is difficult to identify specific contributions from separate factors from the farm environment.” We have also added the following lines to the conclusion section of the paper: “These environmental conversions likely work in concert with other factors such as age, gender, personal hygiene, dietary habits, antibiotic use, and stress to shape the microbiome.” Please see Lines 280-282.
Microbial source tracking While certainly relevant, all these source attribution models whether random forest or the source tracker depends on information regarding as many reservoirs as possible. I am actually a bit surprised that the workers constitute a source of the same since as the pig feces. This surely suggests that other sources should be taken into account.	We wholeheartedly agree with this point. It is likely that there is extensive microbe and gene sharing between different microbial communities on the swine farms. This sharing would extend the possibility that the environmental samples tested are sinks from common sources. Indeed, all of these various sources should be considered possible contributors to the change of students’ gut microbiota. Due to limited sequencing and collection resources, we targeted a subset of the possible sources in this study. Specifically, we considered soil, dust, sewage, pigs, and workers' feces that students are exposed to as potential source environments since several of these samples types have been implicated by prior literature. Importantly, these potential sources also represent a potential intervention point for further large scale epidemiological surveys and resistance gene sharing reduction strategies. To acknowledge this point for the reader, we added the following text on Lines 277-279. “the diversity of source environments we observed in our finite sampling may still only represent a fraction of the true bacterial reservoirs.”
E. coli data This looks very convincing, but is the prevalence of workers and environment really that stable?	For figures 4c and 4d we averaged the occurrence rate for resistance genes for times T1, T2, and T3 within the worker samples and within the environmental samples. To help clarify this we have included the following sentence in the figure legend to indicate that the dotted lines represent sample averages from the three time points. “The dotted lines for worker and environmental samples are the average occurrence rate of resistance genes in E. coli strains from farm workers and environment in T1, T2, and T3.” Please see Lines 341-343.

References

- 1 Falony, G. *et al.* Population-level analysis of gut microbiome variation. *Science* **352**, 560-564, doi:10.1126/science.aad3503 (2016).
- 2 Anderson, M. J. A new method for non-parametric multivariate analysis of variance. *Austral Ecology* **26**, 32-46, doi:10.1111/j.1442-9993.2001.01070.pp.x (2001).
- 3 McArdle, B. H. & Anderson, M. J. Fitting multivariate models to community data: a comment on distance-based redundancy analysis. *Ecology* **82**, 290-297, doi:10.1890/0012-9658(2001)082[0290:Fmmtcd]2.0.Co;2 (2001).
- 4 Smits, S. A. *et al.* Seasonal cycling in the gut microbiome of the Hadza hunter-gatherers of Tanzania. *Science* **357**, 802-806, doi:10.1126/science.aan4834 (2017).
- 5 Li, J. *et al.* Gut microbiota dysbiosis contributes to the development of hypertension. *Microbiome* **5**, 14, doi:10.1186/s40168-016-0222-x (2017).

REVIEWERS' COMMENTS:

Reviewer #1 (Remarks to the Author):

In this revision, the authors have addressed most of my concerns in the previous version regarding the study design and statistical analysis. The revised manuscript has been improved a lot. Now my only concern is that the reproducibility of the study, since the population-level conclusion drawn by the authors is based on a study with a small sample size and only male participants. I think the authors have acknowledged this limitations in the manuscript.

Reviewer #2 (Remarks to the Author):

I believe the manuscript is now suitable for publication.

Reviewer #3 (Remarks to the Author):

I have no further comments. The authors have addressed all the original comments I had.

Response to referees for manuscript NCOMMS-19-13605A, “**Environmental remodeling of human gut microbiota and antibiotic resistome in livestock farms.**” by Jian Sun, Xiao-Ping Liao, Alaric W. D'Souza*, Manish Boolchandani, Sheng-Hui Li, Ke Cheng, José Luis Martínez, Liang Li, You-Jun Feng, Liang-Xing Fang, Ting Huang, Jing Xia, Yang Yu, Yu-Feng Zhou, Yong-Xue Sun, Xian-Bo Deng, Zhen-Ling Zeng, Hong-Xia Jiang, Bing-Hu Fang, You-Zhi Tang, Xin-Lei Lian, Rong-Min Zhang, Zhi-Wei Fang, Qiu-Long Yan, Gautam Dantas, and Ya-Hong Liu.

We thank the referees and others involved in the editorial process, for their time and effort in considering this manuscript and their thoughtful suggestions to help improve it. We have added the limitation of the population-level conclusion in our manuscript and we hope it is suitable for publication in *Nature Communications*.

REVIEWERS' COMMENTS:

Reviewer #1 (Remarks to the Author):

In this revision, the authors have addressed most of my concerns in the previous version regarding the study design and statistical analysis. The revised manuscript has been improved a lot. Now my only concern is that the reproducibility of the study, since the population-level conclusion drawn by the authors is based on a study with a small sample size and only male participants. I think the authors have acknowledged this limitations in the manuscript.

A: Thanks. We added the limitations into the “Discussion” in our manuscript as suggested.

Reviewer #2 (Remarks to the Author):

I believe the manuscript is now suitable for publication.

A: Thanks.

Reviewer #3 (Remarks to the Author):

I have no further comments. The authors have addressed all the original comments I had.

A: Thanks.